# A slow transcription rate causes embryonic lethality and perturbs kinetic coupling of neuronal genes

Magdalena M Maslon[1], Ulrich Braunschweig[2], Stuart Aitken[1], Abigail R Mann[1], Fiona Kilanowski[1], Chris J Hunter[1], Benjamin J Blencowe[2], Alberto R Kornblihtt[3], Ian R Adams[1,*] & Javier F Cáceres[1,**]

## Abstract

The rate of RNA polymerase II (RNAPII) elongation has an important role in the control of alternative splicing (AS); however, the *in vivo* consequences of an altered elongation rate are unknown. Here, we generated mouse embryonic stem cells (ESCs) knocked in for a slow elongating form of RNAPII. We show that a reduced transcriptional elongation rate results in early embryonic lethality in mice. Focusing on neuronal differentiation as a model, we observed that slow elongation impairs development of the neural lineage from ESCs, which is accompanied by changes in AS and in gene expression along this pathway. In particular, we found a crucial role for RNAPII elongation rate in transcription and splicing of long neuronal genes involved in synapse signaling. The impact of the kinetic coupling of RNAPII elongation rate with AS is greater in ESC-differentiated neurons than in pluripotent cells. Our results demonstrate the requirement for an appropriate transcriptional elongation rate to ensure proper gene expression and to regulate AS during development.

**Keywords** ESCs differentiation; kinetic coupling; mouse model; RNA polymerase II; transcription elongation

**Subject Categories** RNA Biology; Transcription

**The EMBO Journal (2019) 38: e101244**

## Introduction

Alternative splicing (AS) is a highly regulated process that generates RNA diversity and is a major contributor to protein isoform diversity. Its regulation depends not only on the interaction of trans-acting factors with regulatory RNA cis-acting sequences but also on multiple layers of regulation, which include DNA methylation, chromatin structure and modification, and transcription (Schwartz & Ast, 2010; Lev Maor *et al*, 2015; Naftelberg *et al*, 2015). The co-transcriptional nature of pre-mRNA splicing led to the suggestion that the rate of transcription elongation acts to control AS in mammalian cells (Beyer & Osheim, 1988; Roberts *et al*, 1998; Pandya-Jones & Black, 2009). Notably, there is a functional relationship between the transcriptional and the splicing machineries, as evidenced by the role of splicing factors, such as TCERG1, also known as CA150 (Suñé & Garcia-Blanco, 1999) and SRSF2 (Lin *et al*, 2008), in stimulating transcriptional elongation. Interestingly, a role for transcription elongation rate influencing splicing fidelity and co-transcriptionality was also observed in yeast (Herzel *et al*, 2017; Aslanzadeh *et al*, 2018).

The elongation control of transcription can be highly regulated and have a profound effect on gene expression. Indeed, following transcription initiation, the transition of RNAPII from a paused to a productive elongation stage constitutes a major rate-limiting step in the transcription of approximately 40% of mRNA-encoding genes (Min *et al*, 2011; Vos *et al*, 2018a,b). Furthermore, transcription elongation is variable, as synthesis rates can differ between genes by several-fold and these variations in elongation rates could be associated with different gene features and epigenetic modifications.

Recent studies revisited the contribution of the kinetics of RNAPII elongation to the regulation of AS, giving rise to two complementary models (Bentley, 2014; Naftelberg *et al*, 2015). The "window-of-opportunity" or kinetic model of AS regulation proposes that the rate of RNAPII elongation influences the outcome of alternative splicing selection. Use of a mutant form of RNAPII (C4/R749H) with a slower elongation rate leads to an increased (de la Mata *et al*, 2003) or decreased (Dujardin *et al*, 2014) inclusion of alternative cassette exons into mature mRNA. A complementary model, termed "Goldilocks", concluded, based on the study of RNAPII mutants with both slow and fast elongation rates, that an

1  MRC Human Genetics Unit, Institute of Genetics and Molecular Medicine, University of Edinburgh, Edinburgh, UK
2  Donnelly Centre, Department of Molecular Genetics, University of Toronto, Toronto, ON, Canada
3  Instituto de Fisiología, Biología Molecular y Neurociencias (IFIBYNE-UBA-CONICET) y Departamento de Fisiología, Biología Molecular y Celular, Facultad de Ciencias Exactas y Naturales, Universidad de Buenos Aires, Ciudad Universitaria, Buenos Aires, Argentina
   *Corresponding author. Tel: +44 131 651 8562; E-mail: ian.adams@igmm.ed.ac.uk
   **Corresponding author. Tel: +44 131 651 8699; E-mail: javier.caceres@igmm.ed.ac.uk

optimal rate of transcriptional elongation is required for normal co-transcriptional pre-mRNA splicing (Fong *et al*, 2014). In both models, recruitment of splicing regulators to cis-acting RNA sequences as well as nascent RNA folding is influenced by the elongation rate of RNAPII (Eperon *et al*, 1988; Buratti & Baralle, 2004; Saldi *et al*, 2018). The global impact of RNAPII elongation rate in the regulation of AS was confirmed with the use of drugs that inhibit RNAPII elongation (Ip *et al*, 2011).

Exogenous agents also affect transcriptional coupling to AS. For instance, UV irradiation promotes RNAPII hyperphosphorylation with the subsequent inhibition of transcriptional elongation, leading to changes in AS, suggesting that transcriptional coupling to AS is a key feature of the DNA-damage response (Muñoz *et al*, 2009; Williamson *et al*, 2017). In plants, light regulates AS through the control of transcriptional elongation by promoting RNAPII elongation, which is negatively regulated in darkness (Godoy Herz *et al*, 2019). To date, all studies investigating the role of transcription elongation in pre-mRNA processing in mammalian systems have been confined to the use of cultured cells transfected with α-amanitin-resistant slow or fast RNAPII elongation mutants. Thus, the consequences of this mechanism of regulation *in vivo* and its effect on cellular differentiation and development remain largely unexplored. Here, we sought to address two important yet largely unexplored questions. First, how does an altered transcriptional elongation rate affect gene expression and the control of AS and impacts on mammalian development? Secondly, what are the extent and the tissue/organism phenotypic consequences of the elongation control of AS? To answer these questions, we generated mouse embryonic stem cells (ESCs) knocked in for a slow RNAPII mutant (C4/R749H). We show that an appropriate RNAPII elongation rate is essential for proper mouse development. We observed that a reduced elongation rate results in major changes in splicing and in gene expression in pluripotent ESCs and along the pathway of neuronal differentiation. The impact of the kinetic coupling of RNAPII elongation rate with AS is more predominant in ESC-differentiated neurons than in pluripotent cells, as it is essential for the expression and splicing of neuron-specific genes involved in synapse signaling.

## Results

### Generation of a slow RNAPII knock-in mutant mouse ES cells

To address the consequences of an altered transcriptional elongation rate for gene expression and for the kinetic control of AS, we set out to generate an *in vivo* model of a slow RNAPII by introducing a heterozygous or homozygous R749H mutation into the endogenous *Polr2a* in mouse ESCs. This mutation is equivalent to the C4 point mutation identified in the *Drosophila* pol II largest subunit, which confers a lower elongation rate, is less capable of transcribing through natural elongation blocks, and causes non-lethal developmental defects in the heterozygous state (Coulter & Greenleaf, 1985; Mortin *et al*, 1988; Chen *et al*, 1996). Gene targeting in mouse ESCs was achieved by rounds of homologous recombination to introduce the R749H mutation into each allele of *Polr2a* to generate heterozygous and homozygous ESCs (Fig 1A, henceforth referred to as WT/slow and slow/slow ESCs).

We verified the correct targeting by PCR of genomic DNA isolated from these ESCs and a diagnostic *Xho*I digest (Fig 1B). Ion Torrent sequencing of overlapping PCR products from ESC genomic DNA encompassing a ~14-kb region around the R749H mutation confirmed that the heterozygous WT/slow and homozygous slow/slow ESCs contained no genomic re-arrangements or additional mutations in this region relative to the parental WT/WT ESCs. We verified the expression of mutant RNAPII in these cells by cDNA sequencing (Fig 1C) and using allele-specific RT–qPCR (Fig 1D).

### Slow transcription elongation hinders early mouse development

The WT/slow ESCs were used to generate a slow RNAPII knock-in mouse model by injection into C57BL/6 blastocysts. We obtained mouse chimeras from these injections; however, no germline transmission was observed upon breeding eight male animals with at least 30% coat color chimerism to C57BL/6 females. These chimeric animals either sired only host blastocyst-derived offspring or were infertile and lacked sperm in the epididymis. As a test, breeding of 3–4 male chimeras is typically sufficient to detect germline transmission (BVAAWF/FRAME/RSPCA/UFAW Joint Working Group on Refinement, 2003). This indicates that ESCs with a heterozygous slow RNAPII appear to be unable to functionally contribute to spermatogenesis. To investigate the developmental consequences of the *Polr2a* R749H mutation further, we set out to generate R749H mutant mice using CRISPR/Cas9 (Fig 2A). Specific single guide RNAs (sgRNAs) against *Polr2a* were microinjected into (C57BL/6 × CBA) F2 zygotes along with the Cas9 mRNA and an oligonucleotide repair template containing the R749H mutation ("slow oligo"), and subsequently, embryos were transferred into pseudopregnant recipient mice at the two-cell stage. No live-borne mice were obtained containing homozygous or even heterozygous mutations in the Polr2a locus among the 47 pups (Fig 2B). To rule out inefficient induction of double-strand DNA breaks (DSBs) by sgRNAs, or inefficient oligonucleotide-mediated repair at this locus, we co-injected the same pair of sgRNAs with a repair template mixture containing a 1:1 ratio of a slow oligo and a silent oligo, the latter being a repair template containing silent mutations. Again, we could not detect the slow mutation in any of the 51 pups born; however, we obtained two homozygotes and four heterozygotes as a result of repair with silent oligo (Fig 2C). Taken together, the ESC chimeras and the CRISPR/Cas9 microinjections suggest that even heterozygosity for *Polr2a* R749H causes developmental defects in mice. Next, we investigated at what stage the *Polr2a* R749H mutation caused embryonic lethality. We microinjected slow oligo along with guide RNAs into zygotes, cultured the zygotes *in vitro* for 3 days, and analyzed the resulting embryos at the late morula/blastocyst stage. We found several slow heterozygous embryos and only 1 homozygous embryo, revealing that the R749H mutation was tolerated at the pre-implantation stage (Fig 2B and C). However, when microinjected zygotes were transferred to pseudopregnant recipient females at the 2-cell stage to allow them to implant and develop further, only one heterozygous and no homozygous slow mutations were found in mid-gestation embryos at E9.5-E11.5 (Fig 2C). Thus, we conclude that the *Polr2A* R749H mutation causes early embryonic lethality.

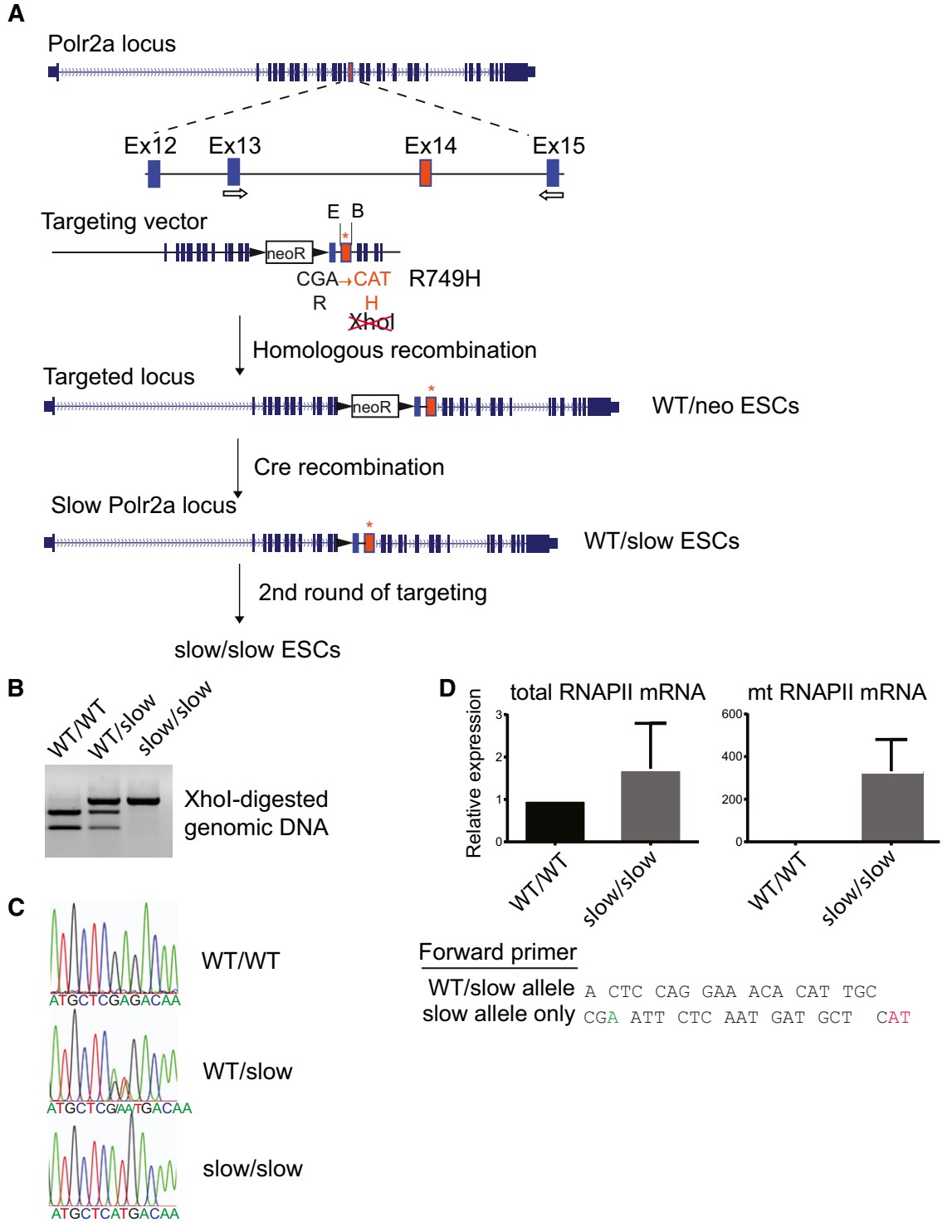

Figure 1. Generation of slow RNAPII knock-in mutant mouse ESCs.

A   Cartoon depicting the mutagenesis strategy, including the genomic target locus, as well as the structure of the targeting vector. Arrows indicate location of primers used for genotyping.
B   Restriction enzyme diagnostic test for the presence of the R749H mutation.
C   Sequence trace of cDNA showing the presence of the heterozygous and homozygous R749H mutations.
D   qRT–PCR with primers specific to both wild-type and mutant RNAPII (left panel) or to the mutant form of RNAPII (right panel), confirming that only the slow version of RNAPII is expressed in homozygous slow/slow ESCs. The sequences of the respective forward primers are shown. The "WT/slow allele" primer is complementary to the sequence in exon 14 upstream of the mutation. The "slow allele only" primer has its 3′ end matching the mutated codon 749 and does not anneal to the WT DNA sequence. The mean ± SEM is plotted, $n = 2$.

**A**

Polr2a locus

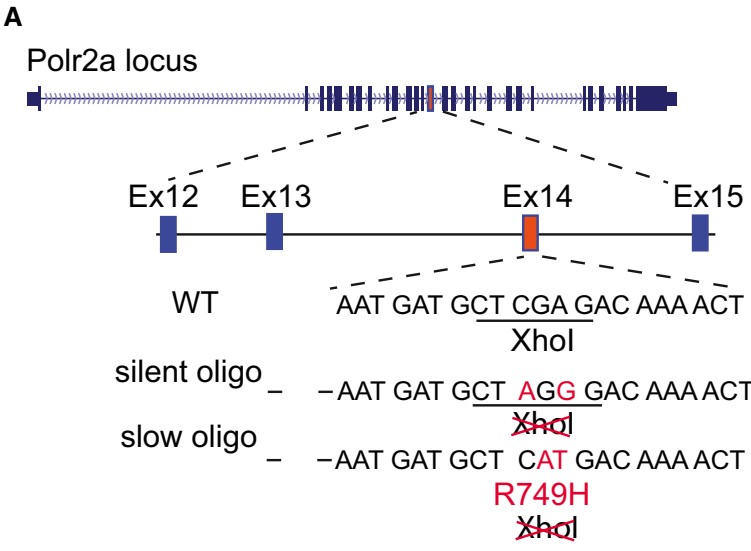

**B**

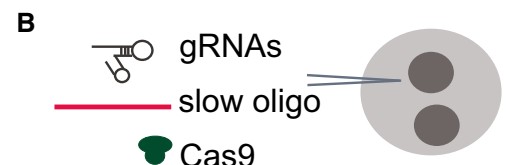

| | Genotype | E3.5 | Born |
|---|---|---|---|
| **WT mice** | WT/WT | 34 | 47 |
| **Slow hets** | WT/slow | 12 | 0 * |
| **Slow homozyg** | slow/slow | 1 | 0 * |

**C**

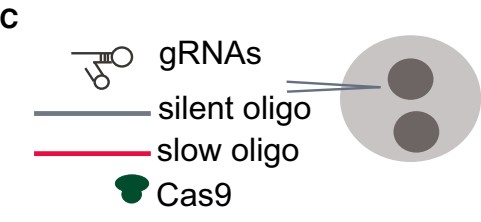

| | Genotype | E3.5 | E9.5 | Born |
|---|---|---|---|---|
| **WT mice** | WT/WT | 86 | 64 | 45 |
| | WT/silent | 7 | 9 | 4 |
| | silent/silent | 4 | 10 | 2 |
| **Slow hets** | WT/slow | 20 | 1* | 0 |
| | silent/slow | 0 | 0 | 0 |
| **Slow homozyg** | slow/slow | 0 | 0 | 0 * |

**Figure 2. CRISPR-/Cas-mediated generation of a slow RNAPII knock-in mutant mouse.**

A    Cartoon depicting the mutagenesis strategy, including the genomic target locus, as well as two repair templates, either introducing a silent mutation (silent oligo) or the R749H mutation (slow oligo). Multiple repair oligo templates were tested with different composition of silent restriction sites.

B, C    The number of embryos/mice of different genotypes recovered after injecting a slow RNAPII repair oligo (B) or a 1:1 mixture of slow and silent RNAP II repair oligos (C) into E0.5 zygotes is shown in each table. The RNAPII genotypes and the stages at which embryos/mice were analyzed are indicated. * indicates $P < 0.01$ (Fisher's exact test relative to E3.5).

## The R749H mutation decreases the transcription elongation rate in mouse ESCs

We analyzed the effect of the slow RNAPII mutation in ESCs using 5,6-dichlorobenzimidazole 1-beta-D-ribofuranoside (DRB) to measure RNAPII transcriptional elongation rates (Singh & Padgett, 2009). DRB inhibits P-TEFb-dependent phosphorylation of the transcription elongation factor Spt5 and of serine 2 in the carboxy-terminal domain (CTD) of RNAPII. Thus, newly initiated RNAPII cannot progress to the elongation phase; however, upon DRB removal, all initiated polymerases are released, and the appearance of selected intron–exon junctions can be monitored by qRT–PCR in a time-dependent manner. We monitored how transcription proceeded through the *Itpr1* and *Utrophin* genes, following DRB removal. Transcription over the first exon–intron junctions did not differ between the wild-type (WT) and mutant cell lines (Fig EV1, Exon 1–Intron 1 panels). However, appreciable pre-mRNA levels at the more downstream exon–intron junctions were detected earlier in WT than in slow/slow cells. For example, the appearance of an exon–intron junction 133-kb downstream from the *Itpr1* transcription start site was detected at 40 min post-DRB release for the WT, as compared to 90 min for the mutant RNAPII (Fig EV1A, see Exon 5–Intron 5 panel). An overall mean elongation rate across *Itpr1* and *Utrophin* was estimated to be 3.3 and 5.6 kb/min, respectively, in WT cells, as

compared to 1.5 and 1 kb/min in slow/slow cells. We also measured overall transcription using a reversible DRB block followed by incubation with medium containing tritiated ³H-uridine. Time-resolved accumulation of newly made RNA, as measured by the incorporation of ³H-uridine, was attenuated in slow/slow in comparison with WT ESCs (Fig EV1B). We also found that nuclear extracts isolated from slow/slow cells were less efficient in driving the production of a runoff transcript from the artificial DNA template (Fig EV1C). These results are in agreement with the previous observation that the R749H mutation in RNAPII leads to approximately a twofold decrease in the transcription elongation rate *in vitro* (Boireau *et al*, 2007) and that the elongation rate positively correlates with expression levels (Danko *et al*, 2013; Jonkers *et al*, 2014).

Next, we analyzed RNAPII elongation rates genome-wide using metabolic labeling of newly transcribed RNAs by the uridine analogue, 4-thiouridine (4sU) (Rädle *et al*, 2013; Fuchs *et al*, 2014). Transcription was arrested with DRB for 3 h; then, DRB was removed and cells were allowed to transcribe for 5 and 15 mins. To label the newly transcribed RNA, cells were pulsed with 4sU for the last 10 min of each time point (Fig 3A). Cells not released from transcriptional block ("0 min") were also labeled with 4sU. Following biotinylation and purification, 4sU-labeled RNAs were subjected to deep sequencing. At time "0 min", which corresponds to the release from DRB inhibition, the vast majority of reads were observed over a narrow area near the promoter (Fig 3B, black line, and Fig EV2A, top panel). As time progresses, the reads from nascent RNA are observed further into the gene bodies, referred to as the transcription "wave-front" progression (Fig EV2A, wave-front progression in *Notch1*). On average, we observed that in WT cells, RNAPII had progressed approximately 11 kb into the gene at 5 min and up to 35.8 kb after 15 min after DRB removal. By contrast, in slow/slow cells the transcription wave-fronts reached only 8.6 kb and 26.7 kb at 5- and 15-min time points, respectively (Figs EV2B and 3B). Genome-wide, we observed an average elongation rate of 2,450 bases/min in wild-type cells, but reduced rates of 1,780 bases/min in slow/slow cells (Fig 3B and D). Previous work suggests that the speed of RNAPII differs between genes (Danko *et al*, 2013; Jonkers *et al*, 2014). The density plot of reported elongation rates demonstrates that the dynamic range of transcription rates is narrower in slow RNAPII cells, while in wild-type cells it seems to be bimodal, revealing a population of RNAPII transcribing at higher rates (Fig 3C). Indeed, most genes have a lower elongation rate in slow/slow cells in comparison with wild-type cells (e.g., *Ern1* is transcribed at 4.2 and 1.9 kb/min, in wild-type and slow/slow cells, respectively (Table EV1). Interestingly, there are examples of genes that are transcribed faster in slow/slow cells. It is possible that a slower elongation rate might lead to a longer residence time, allowing more time for positive factors to bind and/or stimulate RNAPII and consequently lead to overall higher transcription rates for these genes. Finally, there is a positive correlation between elongation rate and expression levels (Fig EV2C), indicating that on average, highly expressed genes have faster elongation rates in agreement with previous reports (Danko *et al*, 2013; Jonkers *et al*, 2014). Overall, these data validate previous results obtained in cultured cells transfected with an α-amanitin-resistant RNAPII harboring the C4 mutation (de la Mata *et al*, 2003; Fong *et al*, 2014) and confirms that the endogenous knock-in of a slow RNAPII mutation affects negatively the transcriptional elongation rate in mouse ESCs.

## Role of transcriptional elongation during neural differentiation

To assess whether a differential transcription elongation rate affects ESC differentiation, we exploited an *in vitro* model of neuronal development. During embryonic development, different pathways control self-renewal and differentiation capacity of neural progenitors (Doe, 2008; Aguirre *et al*, 2010). ESCs can differentiate into multipotent Sox1 and Nestin-positive neuronal progenitor cells (NPCs) in a serum-free adherent monolayer culture (Ying *et al*, 2003; Conti *et al*, 2005) (Fig 4A). The ESC-derived NPCs can then be used to generate neural stem cells (NSCs) by allowing these cells to form floating aggregates (AGGs) in epidermal/fibroblast growth factor 2 (EGF/FGF2)-containing medium from which a population of bipolar, self-renewing and multipotent NSCs outgrow in adherent conditions (Fig 4A). Alternatively, NPCs can be differentiated into all three neural lineages. For example, when cultured adherently on poly-ornithine/laminin in media containing cAMP and ascorbic acid, they differentiate into Tuj1+ immature neuronal cells and further into Map2-positive mature post-mitotic neurons.

We induced differentiation of WT ESCs and slow/slow ESCs into NPCs. We found that both wild-type and slow/slow cells generated Sox1, Pax6, and Nestin-positive NPCs (Fig 4B); however, we also observed decreased proliferation or compromised differentiation potential of slow/slow cells (see Materials and Methods). We next tested whether we could generate NSCs from slow/slow ESC-derived NPCs. Interestingly, we found that despite obtaining neural AGGs (Fig 4C), slow/slow NSCs could not be maintained in EGF-/FGF2-proliferating conditions (Fig 4D). Instead, following a few passages we noted the appearance of flattened differentiated cells in the slow/slow cultures, and subsequently, we observed overwhelming cellular death. Strikingly, among some of the remaining Nestin-positive cells in these slow/slow cultures, we observed promiscuous differentiation to Tuj1+ cells (Fig 4D). These results suggest that the balance between maintenance of the self-renewing cell state and differentiation might be perturbed in slow/slow NSCs.

Indeed, Gene Ontology (GO) analysis revealed that those genes upregulated in slow/slow NPCs and in aggregates (AGG) were involved in neuronal functions (Table EV2 and Appendix Fig S1A), which might explain some of the phenotypes observed in slow/slow NSCs (Fig 4). We observed upregulation of *Ascl1*, *Nr2f1*, *Crabp2*, and *Nr6a1* genes (Appendix Fig S1B) in slow/slow NPCs and AGGs. Their overexpression has been previously shown to suppress proliferation of progenitor cells, induce neurogenesis and neuronal maturation (Chanda *et al*, 2014; Gkikas *et al*, 2017), and could explain the premature differentiation observed in slow/slow NSCs. In parallel, we observed that the EGF receptor (EGFR) was twofold downregulated in slow/slow NPCs (Table EV2). As EGF withdrawal causes massive cell death and premature differentiation observed in slow/slow NSCs (Conti *et al*, 2005), decreased expression of EGFR in slow/slow NPCs could contribute to the observed lack of their self-renewal in EGF-/FGF2-proliferating conditions. Although the slow RNAPII allele appears to impair the maintenance of NSCs, the presence of differentiated Tuj1 neurons in the NSC cultures (Fig 4D) suggests that a slow transcriptional elongation rate does not impair neuronal differentiation per se. Indeed, when re-plated onto poly-ornithine-/laminin-coated plates, both WT and slow/slow NPCs,

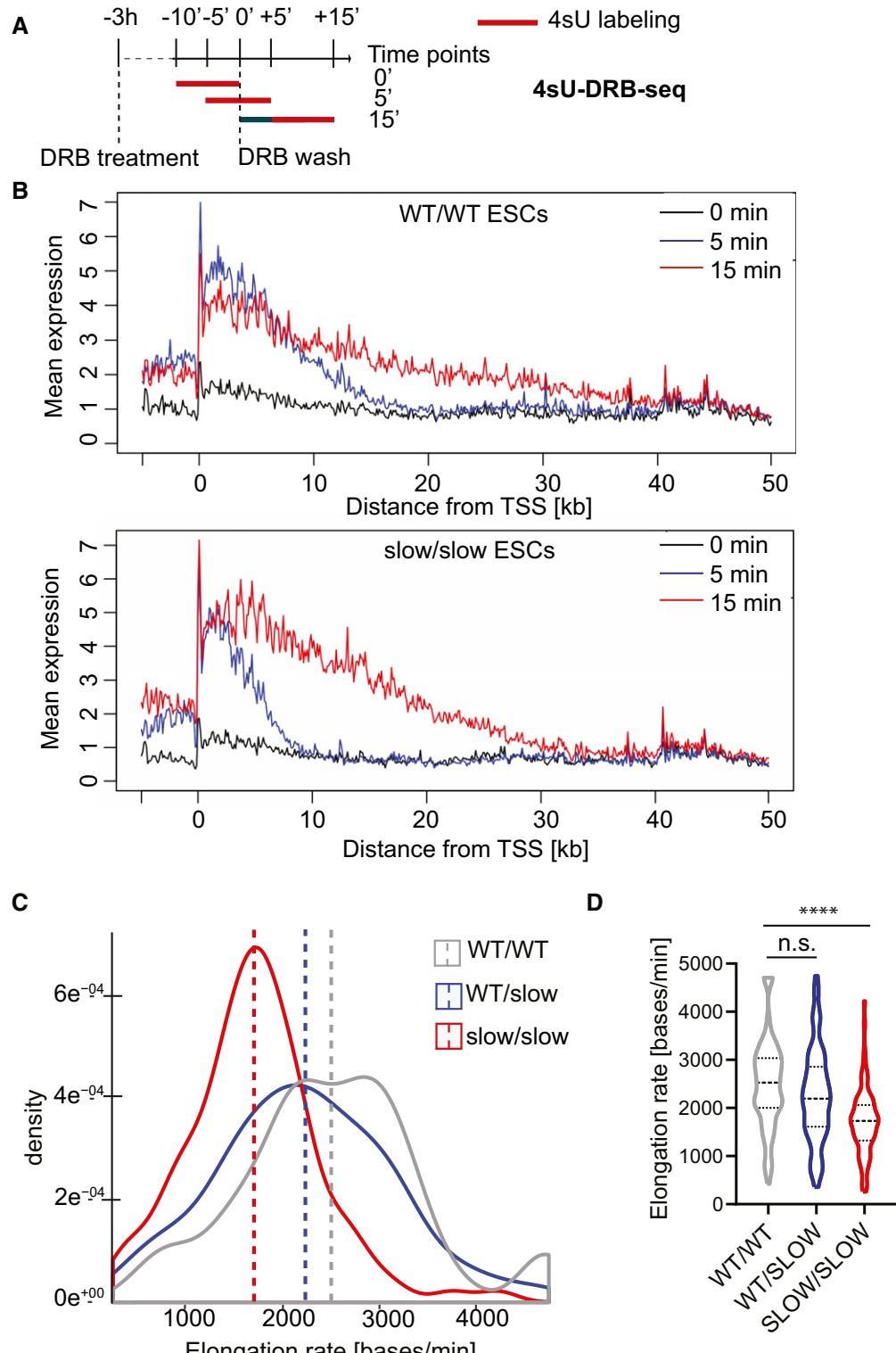

**Figure 3. Global analysis of transcription elongation rate in mouse ESCs by 4sU-DRBseq.**

A       Schematic of the 4sU-DRB-seq labeling protocol.

B       Meta-gene profile of normalized 4sU-DRB-seq reads in WT/WT and slow/slow ESCs.

C, D    Density and violin plot of elongation rate (bases/min) calculated for genes common in all genotypes in WT/WT, WT/slow, and slow/slow ESCs. Box and whisker plot
        (5th–95th percentile) indicates median. Mann–Whitney test, ****$P < 0.0001$.

Source data are available online for this figure.

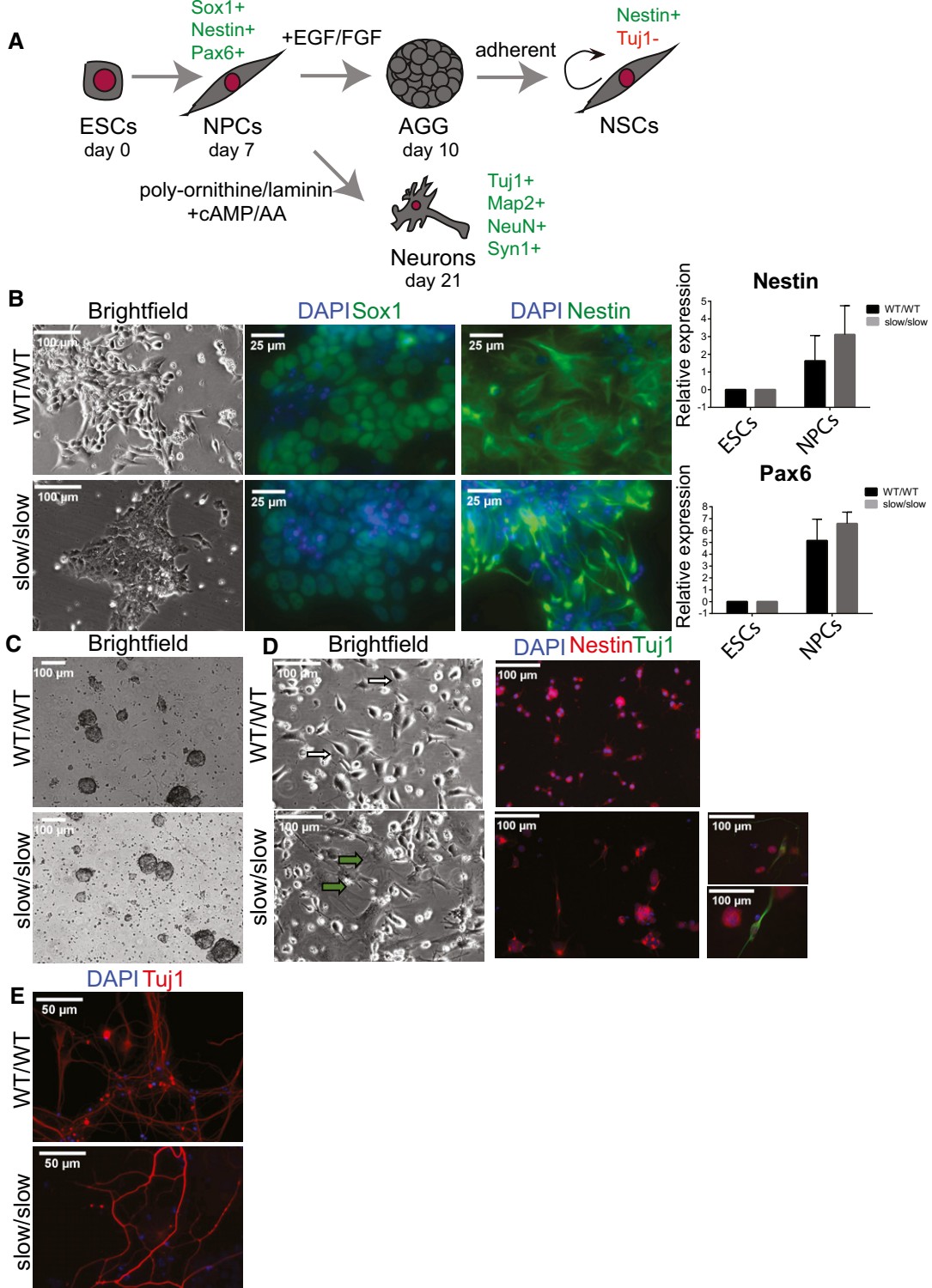

**Figure 4.  Differentiation of WT and slow ESCs along the neural lineage.**

A   Schematic of the neural differentiation system used in this study, indicating the relevant markers that define different stages of differentiation.

B   Bright-field images and analysis of NPC markers by immunofluorescence staining (Sox1 and Nestin) or RT–qPCR (Nestin and Pax6) ($n$ = 3, mean $\pm$ SEM).

C   Bright-field images of aggregates.

D   Bright-field images and immunofluorescence staining for Nestin and neuronal marker Tuj1 in NSC cultures grown in EGF-/FGF-proliferating conditions. White arrows indicate NSCs and green arrows differentiated cells. Two small panels on the right are examples of Tuj1+ neuronal cells in slow/slow NSC cultures.

E   Immunofluorescence staining for neuronal marker Tuj1 in neuronal cultures grown on poly-ornithine/laminin at 21 days of differentiation.

differentiated into Tuj1-, Map2-, and NeuN-positive neurons (Figs 4E and EV3A and B). Whereas we observed a robust expression of the synaptic marker (Syn1) in WT neurons, it seemed reduced in slow/slow neurons (Fig EV3A and B). Overall, these data show that the slow mutation in RNAPII causes problems in the maintenance/self-renewal of NSCs but appears not to interfere with neuronal differentiation *per se*. It also suggests that neurons harboring a homozygous slow mutant RNAPII might be functionally or developmentally different than WT neurons.

## Transcriptional elongation rate influences alternative splicing decisions in ESCs and during neural differentiation

Next, we investigated gene expression and AS changes by RNA sequencing (RNA-seq) analysis of poly (A)+ RNA isolated from pluripotent ESCs, NPCs, and neurons. First, we compared alternative exons' usage between wild-type and slow/slow cells using vasttools, which assigned a "percentage spliced in" (PSI) value to each exon. Analysis of AS changes revealed 75, 167, and 415 events of enhanced exon inclusion, comprising cassette exons and microexons, in slow/slow ESCs, NPCs, and neurons, respectively, as compared to their WT counterparts (Fig 5A and Table EV3). We also observed that whereas cassette exon events did not show a bias toward increased exon inclusion in slow/slow ESCs or NPC cells when compared to WT cells, there was some tendency for an increased exon inclusion in neurons (60% of alternative cassette exons are more included in slow/slow neurons) (Fig 5A). By contrast, we found that exon skipping was enhanced relative to cassette inclusion by the slow RNAPII mutant, with 91 and 510 skipped cassette exons and microexons, detected in ESCs and NPCs, respectively. This is compatible with the current models of kinetic coupling, where a slow RNAPII can lead either to enhanced exon inclusion if the AS event depends on the recruitment of positive regulators or to exon skipping if splicing inhibitors are recruited (Dujardin *et al*, 2014).

The splicing signature in slow/slow cells could be a direct result of perturbations in the elongation rate or be due to an indirect effect through changes in expression of splicing factors and/or RNA-binding proteins (RBPs). Indeed, we found that, for example, Mbnl2 was downregulated in ESCs and NPCs, whereas Nova1 was downregulated in neurons (Appendix Fig S2A). We also used available datasets from experimental perturbations of some of these differentially expressed splicing factors and found alternatively spliced mRNAs that were targets of these differentially expressed RBPs (Appendix Fig S2B). However, there was not a significant difference in the proportion of such indirect events between ESCs and differentiated cells, suggesting that most of the events differentially identified in this study correspond to events directly affected by the rate of elongation of RNAPII. Importantly, the extent of splicing changes was much more pronounced in NPCs and fully differentiated neurons in comparison with ESCs (166 cassette exons and microexon events changing in slow/slow ESCs, as compared to 677 or 693 cassette exons and microexons changes observed in slow/slow NPCs and neurons, respectively, Fig 5A). We validated a selected number of alternatively spliced events by RT–PCR analysis (Fig 5B and Appendix Fig S3). Given that the total number of detected AS events in the different stages of neuronal differentiation is comparable (Fig 5A and B), these results underscore the increased

importance of kinetic coupling as differentiation progresses. A possible explanation for this observation is related to changes in chromatin structure during cell differentiation. Chromatin is reported to be more open and accessible in pluripotent ESCs (Gaspar-Maia *et al*, 2011). This differential chromatin organization will likely have a direct influence in the elongation rate of RNAPII (Selth *et al*, 2010; Naftelberg *et al*, 2015). Not only did we observe an elevated number of affected exons in slow/slow neurons in comparison with slow/slow ESC, but also the number of splicing changes increases during differentiation to neurons, with 1,365 alternative splicing events detected in WT cells upon differentiation, whereas this number increases to 2,252 exons in slow/slow cells (Table EV3). We examined the properties of elongation rate-sensitive exons, namely 5′ and 3′ splice sites strength, as well as the length of flanking introns and alternative exon (Yeo & Burge, 2004; Corvelo *et al*, 2010) (Fig EV4A). We noted that exons that were more included in slow/slow ESCs had longer flanking introns (median of 2,335 and 1,546 bases in included and not-affected exons, respectively). Consistent with the "window-of-opportunity" model of kinetic coupling, these longer introns could contribute to a time delay significant enough to promote recognition and splicing of suboptimal exons in nascent transcripts. By contrast, exons affected in slow/slow neurons did not show such characteristics and seemed to be more dependent on the repertoire of expressed RBPs. For example, RNA maps produced for RNA-binding proteins (CISBP-RNA IUPAC-binding motifs; Ray *et al*, 2013) revealed that introns downstream of exons skipped in slow/slow neurons are enriched for Nova1-binding sites (Fig EV4B), and indeed, binding of this factor downstream of alternative exons has been previously shown to enhance their exclusion (Ule *et al*, 2006). Conversely, we noted increased occurrence of Rbfox1-binding motifs in the introns downstream of exons showing more skipping in slow RNAPII-expressing neurons (Zhang *et al*, 2008). As the levels of Rbfox1 remain the same between WT and slow/slow neurons, this observation indicates some functional connection between this splicing factor and kinetic coupling.

High-throughput RNA-seq of poly (A)+RNA revealed changes in the expression of several hundreds of genes in slow/slow cells, as compared to their WT counterparts (Appendix Fig S4A and B, and Table EV2). We, therefore, looked at whether the observed changes in AS are coupled to changes in the expression of corresponding genes. Notably, the differential splicing observed in the presence of a slow elongating RNAPII is generally not driven by differential gene expression (Fig EV5A–C). The only exception are some cases of intron retention, where negative correlation with the expression might reflect frequent coupling of intron retention to NMD (Fig EV5A–C). Thus, we conclude that the majority of AS changes are not merely a consequence of a differential gene expression between ESCs, NPCs, and neurons, but rather show specific sensitivity to RNAPII speed during differentiation (Appendix Table S1, Fig EV5).

## Slow transcription elongation perturbs expression of long synaptic genes

Enrichment Map visualization of gene sets enriched among downregulated and alternatively spliced genes in slow/slow neurons revealed that they are involved in programs that are

**A**

| | ESCs | | | NPCs | | | Neurons | | |
|---|---|---|---|---|---|---|---|---|---|
| | UP | DOWN | Total | UP | DOWN | Total | UP | DOWN | Total |
| **Cassette exon** | 73 | 81 | 22391 | 160 | 473 | 24065 | 402 | 224 | 24823 |
| **Microexon** | 2 | 10 | 598 | 7 | 37 | 678 | 13 | 54 | 692 |
| **Alt 3'/5' splice site** | 19 | 17 | 18915 | 156 | 153 | 20422 | 62 | 69 | 20896 |
| **Intron Retention** | 32 | 10 | 23421 | 151 | 98 | 26253 | 127 | 81 | 26910 |
| **ALL** | **126** | **118** | **65325** | **474** | **761** | **71418** | **604** | **428** | **73321** |

**B**

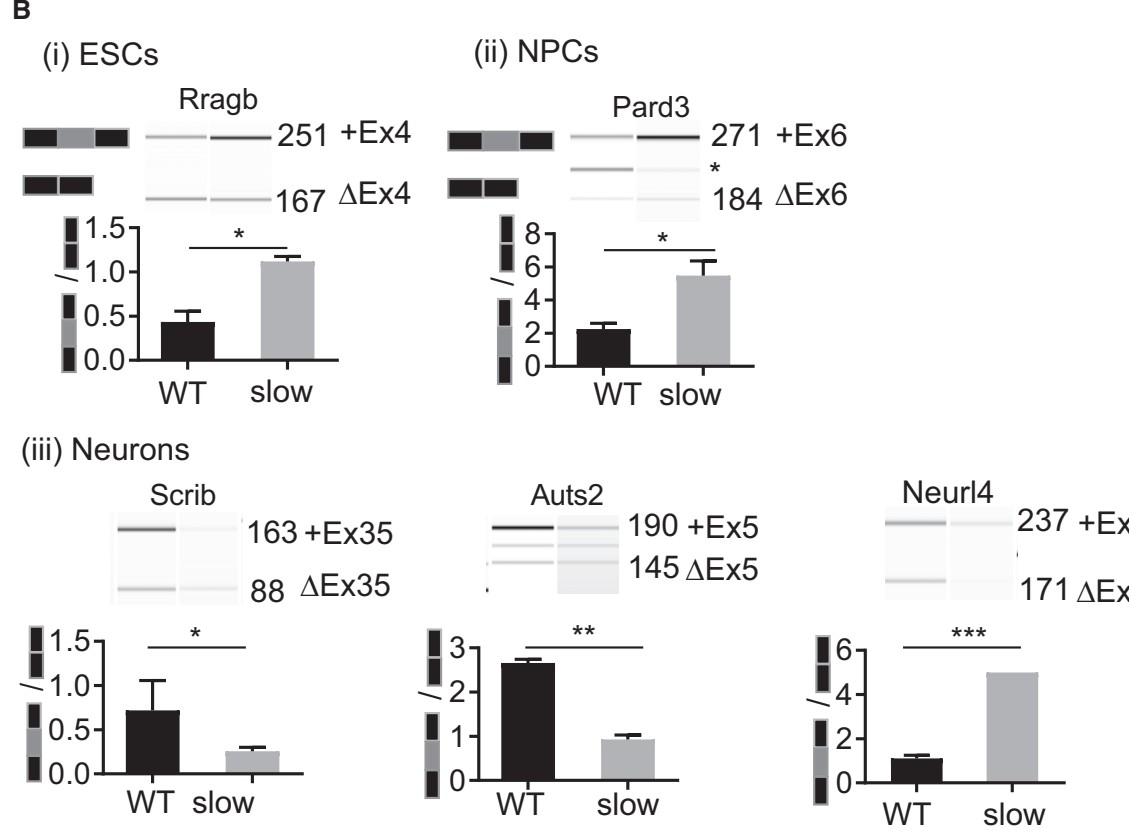

**Figure 5. The rate of transcriptional elongation influences alternative splicing decisions in ESCs and during neural differentiation.**

A   Number of alternative splicing events that are sensitive to a slow elongation rate, including cassette exons, microexons, alternative 3′ or 5′ splice sites, and retained introns in ESCs and at different stages of neural differentiation. UP and DOWN refer to increased or decreased levels of a splicing event in slow/slow cells relative to WT cells with dPSI (percent spliced in) ≥ 10% and FDR < 0.05 when comparing regulated events to all detected events and retained (Total). Splicing was quantified using VAST-TOOLS.

B   RT–PCR analysis validation of selected alternatively spliced exons. RT–PCR was performed on total RNA from WT or slow/slow ESCs, NPCs, or neurons. PCR products were visualized and quantified by Bioanalyzer (Agilent). Images are representative of experiments performed in triplicate. The mean ± SEM is plotted with *P < 0.05, **P < 0.005, ***< 0.0001 as determined by t-test.

Source data are available online for this figure.

essential for synapse formation and synaptic signaling (Fig 6A and Appendix Fig S5). Indeed, genes downregulated in slow/slow neurons encode proteins involved in the entire life cycle of synaptic vesicles: among them are Syn1 and Syn2, which tether the vesicle to the actin cytoskeleton (Thomas *et al*, 1988); Snap25, Stx1b, Stxbp1, and Syt1 proteins, which are involved in

synaptic vesicle fusion and recycling; and neurexins (including Nrxn1 and Nrxn2) and contactin-associated proteins (including Cntnap2 and Cntnap3) that form the synaptic scaffolding system and are involved in trans-synaptic communication. Similarly to downregulated genes, alternative splicing events involved proteins that are important for synaptic signaling (Table EV4). For

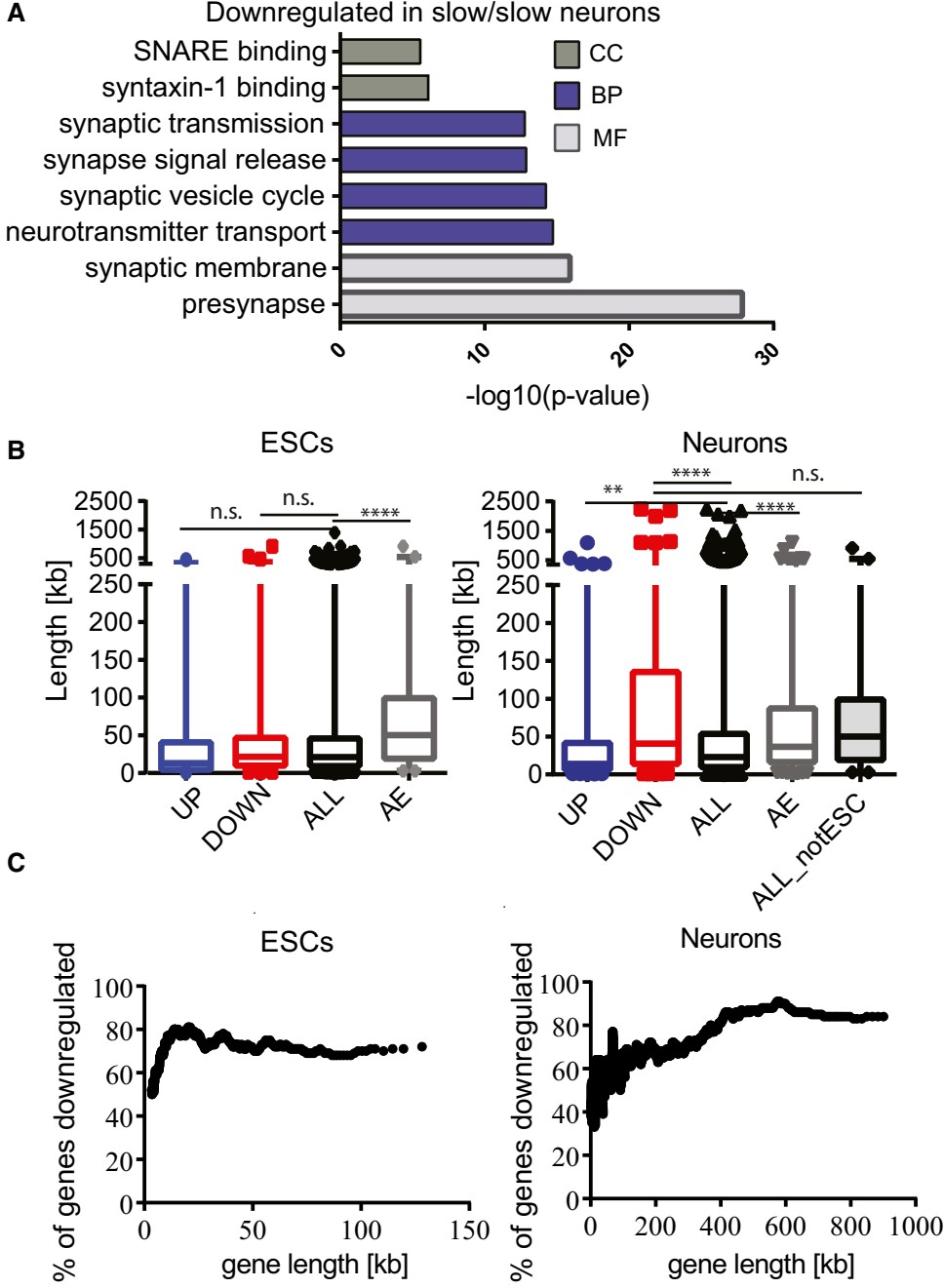

**Figure 6. A slow RNAPII preferentially affects synaptic genes.**

A   Gene ontology analysis of genes downregulated in slow/slow neurons showing the top-ranked cellular component (CC), biological processes (BP), or molecular functions (MF) GO categories.

B   Box plot showing the length of upregulated (UP), downregulated (DOWN), not affected genes (ALL), and pre-mRNAs affected by alternative splicing (AE, for alternative exons) for ESCs (left panel); length of upregulated (UP), downregulated (DOWN), not affected genes (ALL), pre-mRNAs affected by alternative splicing (AE, for alternative exons), and not affected genes that are only expressed in neurons (but not in ESCs) (ALL_notESC), for neurons (right panel). n.s., $P > 0.05$, **$P < 0.005$; ****$P < 0.0001$ as determined by Mann–Whitney $t$-test. Boxes delimit the first and third quartiles. The horizontal lines represent the data medians. Whiskers are drawn down to the 1st and 99th percentiles.

C   Percentage of genes that are downregulated in ESCs and neurons, plotted as a sliding window of 100 genes by length.

example, we observed increased skipping of alternative exons in *Scrib*, a gene encoding a protein involved in neurotransmitter release (Fig 5B). We also confirmed differential splicing of Exon

7 in Apbb2, a protein involved in synaptic vesicle loading (Appendix Fig S3). We noted altered splicing among members of neurexins, synaptic receptors that undergo an extensive

combinatorial use of AS to provide molecular diversity required for the functional differentiation of synapses (Table EV3) (Schreiner *et al*, 2014). Finally, we observed AS events in proteins involved in the synaptic vesicle cycle, including both pre-synaptic and post-synaptic space, among them Stx4A, Syn1, Synj1, Stx3 and many others. Some of these AS events result in premature termination codons, others change domain structures or affect ion transfer, hence all likely contribute to the function or the specificity of the synapse (Table EV4).

Interestingly, we noticed that those genes that are preferentially downregulated in slow/slow neurons are significantly longer than those that were not affected or that are upregulated (Fig 6B, right panel). By, contrast, we found no significant change in the average gene length of downregulated genes in ESCs (Fig 6B, left panel). Notably, a slow transcriptional elongation rate reduced expression of nearly all long genes in neurons, with the percentage of downregulated genes in slow/slow neurons progressively increasing from around 40% for 10 kb genes to over 80% for extremely long genes (Fig 6C, right panel). Some examples of such genes include *Cntnap2* (2.25 Mb) and *Nrxn1* (1.05 Mb) (Table EV2). In contrast, ESCs do not express such long genes (Fig 6C, left panel) and we observe a similar effect caused by a slow elongation across the entire range of gene lengths. From this, we speculate that an optimal elongation rate is important to sustain transcription and splicing of particularly long transcripts that are required for neuronal function. Indeed, recent reports propose that long genes require special mechanisms to specifically maintain long-distance transcription. As an example, the neuronal RNA-binding protein Sfpq (proline-/glutamine-rich, also known as PSF) has been shown to be a critical factor for maintaining transcriptional elongation of long genes (Patton *et al*, 1993; Takeuchi *et al*, 2018).

Thus, we found that both downregulated and preferentially alternatively spliced genes in slow/slow ESC-derived neurons converge onto long genes that are involved in synaptic function. Candidate genes involved in neurodevelopmental diseases encode synapse proteins and are exceptionally long (Bourgeron, 2015). We identified synapse signaling as a major pathway downregulated and mis-spliced in slow/slow neurons and found that slow RNAPII downregulated almost all long genes in neurons. Therefore, we further analyzed the overlap of the genes downregulated and differentially spliced in slow/slow neurons with available datasets for brain disease, including causative genes for autism and schizophrenia (SFARI). We noted that genes differentially expressed and spliced in slow/slow neurons significantly overlapped with those linked to neurological disorders, including ASD disease (Appendix Table S2). From these experiments, we can conclude that a reduced transcriptional elongation rate preferentially affects the expression and alternative splicing of long synaptic genes.

In summary, the development of a genetic system based on knock-in for a slow RNAPII mutation in mouse ESCs unequivocally established that an appropriate RNAPII elongation rate is essential for proper mouse development and for gene expression and its kinetic coupling with AS. Interestingly, the kinetic control of AS is predominantly affected in differentiated cells, suggesting that the chromatin environment represents an important determinant of this coupling. Altogether, our results provide compelling evidence that

transcription elongation rates can have a regulatory role in the expression of genes and the regulation of their alternative splicing patterns during development.

## Discussion

### A slow elongation rate causes early embryonic lethality

We observed that a slow RNAPII mutant caused embryonic lethality even in heterozygosity (Fig 2). There is evidence that the transcriptional output is crucial in specific developmental stages associated with stem cell expansion, as evidenced by their hypertranscription states (Koh *et al*, 2015; Percharde *et al*, 2017). It is possible that a slow elongation rate cannot sustain the high levels of mRNA production required at early stages of development. It was suggested that progenitor cells might also require hypertranscription to allow for their expansion. Similarly, loss of self-renewal in slow/slow NSCs could be related to the inability of cells harboring a slow RNAPII to maintain the required levels of transcriptional output. Initial stages of mouse embryonic development display a great range of cell cycle duration, from up to 20 h for the first cell division to 2- to 3-h cell cycles during gastrulation (Artus *et al*, 2006) or 8 h during initial stages of murine neurogenesis (Takahashi *et al*, 1995). In this scenario, a reduced elongation rate in slow/slow mutant embryos might not allow efficient transcription or might delay expression of some crucial mRNAs that need to be expressed in these fast dividing cells. Whereas in mice both the homozygous and heterozygous slow mutations result in embryonic lethality (Fig 2), the C4 mutation in *Drosophila* is tolerated in heterozygosity where the flies present a mutant phenotype called "Ubx effect" that resembles the one seen in flies haploinsufficient for the Ubx protein. This was attributed to Ubx mis-splicing as it is one of the few *Drosophila* genes with an extremely long intron (50 kb) (de la Mata *et al*, 2003). This is in agreement with results presented here showing that mouse genes with long introns are preferentially affected by a slow RNAPII in ESC-differentiated neurons.

### Kinetic coupling is enhanced in neurons

We found that the impact of RNAPII elongation rate on AS is predominant in ESC-differentiated NPCs and neurons. This is most likely caused by a distinct chromatin environment between pluripotent and differentiated cells having a differential impact on RNAPII transcriptional elongation rate, since previous evidence indicated that the C4 mutation is not catalytically slow, but rather less efficient in overcoming internal pauses (Chen *et al*, 1996). While chromatin is quite dispersed in E3.5, heterochromatin foci appear in E5.5, which corresponds to the epiblast stage following embryo implantation (Ahmed *et al*, 2010). Indeed, despite the conflicting literature regarding deposition of histone marks throughout differentiation (Azuara *et al*, 2006; Wen *et al*, 2009; Lienert *et al*, 2011), a large body of evidence suggests that chromatin undergoes dynamic changes during differentiation leading to a more compact environment in the differentiated state. Various mechanisms might promote a switch from a more open to a more compact chromatin state during cell differentiation,

including an increase in repressive histone marks, a local change in nucleosome occupancy, or a general increase in histones' levels (Fiszbein *et al*, 2016; Gavin *et al*, 2017; Yoon *et al*, 2018). It was reported that the nuclei of ESCs macroscopically appear to contain less condensed chromatin, whereas well-defined foci of compact heterochromatin become evident in ESC-derived NPCs (Meshorer *et al*, 2006). Indeed, chromatin structure can become a major impediment to transcriptional elongation and histone modifications can directly affect the nucleosomes, by either loosening or tightening DNA binding around them (Veloso *et al*, 2014; Jonkers & Lis, 2015). Moreover, exons have a negative effect on RNAPII elongation rate, which could be associated with exonic features, such as a higher CG content, and exon-specific histone marks (H3K36me3 and H3K4me1) (Jonkers *et al*, 2014). An example of a crosstalk between the chromatin environment and AS has been shown in the case of exon 18 in the neural cell adhesion molecule (NCAM), where membrane depolarization of neuronal cells induces a local H3K9 hyperacetylation, resulting in exon skipping (Schor *et al*, 2009). Conversely, inducing a more compact chromatin state by transfection of siRNAs targeting the intron downstream of an alternative exon promotes H3K9 and H3K27 methylation, HP1 recruitment, in turn leading to local roadblocks for RNAPII elongation rate and increased kinetic coupling (Alló *et al*, 2009). We speculate that the specific changes in chromatin structure during differentiation might create natural "roadblocks" to elongating RNAPII, which is further enhanced in slow RNAPII-expressing cells leading to increased kinetic coupling observed in NPCs and neurons derived from slow/slow cells.

### An appropriate elongation rate sustains expression and splicing of long genes involved in synapse signaling in neurons

Slow RNAPII leads to specific downregulation of longer genes in neurons. Intriguingly, neurons express the longest genes among different cell types and many of these encode proteins involved in neuronal development and synapse formation. As such, a slow elongating RNAPII could preferentially affect transcription and splicing of those long genes. Dysregulation of the expression of these long genes might represent a mechanism underlying neurodegenerative and psychiatric disorders (King *et al*, 2013;

Gabel *et al*, 2015). For example, loss of FUS/TLS and TDP43, genes linked to ALS, preferentially affects splicing of long pre-mRNAs (Lagier-Tourenne *et al*, 2012). The neuronal RBP SFPQ, which is required to sustain long-distance transcription elongation for longer genes (Takeuchi *et al*, 2018), has a role in neurodegenerative diseases, including ALS, ASD, and frontotemporal lobar degeneration (FTLD). The differentially expressed and alternatively spliced pre-mRNAs in neurons are involved in synapse signaling, neurite outgrowth, and axonal guidance. We speculate that mis-regulation of RNAPII elongation rate could have detrimental implication in neurodevelopment, preferentially affecting the expression and/or splicing of synaptic proteins, which are encoded by particularly long genes (Fig 6). Indeed mutations in genes involved in synaptic signaling lead to neurodevelopmental diseases, including autism spectrum disorder (ASD) (Bourgeron, 2015). Importantly, chromatin remodeling, transcription, and splicing genes have been identified in genetics studies of *de novo* mutations in autistic patients (De Rubeis *et al*, 2014). These genes encode proteins that are active during brain development and are important in transcription elongation, either through direct interaction with RNAPII or through indirectly affecting chromatin structure. Physiological conditions that could alter RNAPII elongation or mutations disrupting elongation rate control might affect preferentially the nervous system, as these cells express particularly long genes. This could not only result in changes in transcription but also affect AS patterns via kinetic coupling. These observations highlight an essential role for an appropriate elongation rate in gene expression and splicing regulation during neural development and suggest that its mis-regulation could underlie some neurological disorders.

In conclusion, we show that a slow elongation rate affects gene expression and AS, consistent with the coupling of transcription with splicing. This kinetic control of AS is more strongly affected as differentiation progress. Most notably, we identify elongation rate control as a major mechanism to sustain transcription and splicing of long neuronal genes involved in synapse signaling. This study provides a compelling evidence that transcription elongation rates have a regulatory role in the expression of genes and the regulation of their AS patterns during development.

# Materials and Methods

### Reagents and Tools table

| Reagent | Manufacturer | Catalog number |
| --- | --- | --- |
| **Antibodies** | | |
| Tuj1 | BioLegend | 801213 |
| Map2 | Millipore | MAB3418 |
| RNAPII (8WG16) | Abcam | ab817 |
| Syn1 | Novus Biologicals | NB300-104 |
| Tubulin | Sigma | T9026 |
| NeuN | Abcam | Ab177487 |
| Nestin | Abcam | Ab24692 |

**Reagents and Tools table** (continued)

| Reagent | Manufacturer | Catalog number |
|---|---|---|
| **Chemicals** | | |
| 5,6-dichlorobenzimidazole 1-β-d-ribofuranoside (DRB) | Sigma-Aldrich | D1916 |
| 4-thiouridine | Sigma-Aldrich | T4509 |
| EZ-Link™ HPDP-Biotin | Thermo Scientific | 21341 |
| Dimethylformamide | Thermo Scientific | 20673 |
| No-Weigh dithiothreitol (DTT) microtubes | Thermo Scientific | 20291 |
| **Cell culture reagents, cytokines** | | |
| Recombinant Murine EGF | Peprotech | 315-09 |
| Recombinant Human FGF-basic | Peprotech | 100-18B |
| Ascorbic Acid | Stemcell Technologies | 07157 |
| Dibutyryl-cAMP | Stemcell Technologies | 73886 |
| DMEM/F12 | Thermo Scientific | 31331028 |
| Neurobasal medium | Thermo Scientific | 21103049 |
| GMEM | Thermo Scientific | |
| MEM Non-essential amino acid | | |
| Sodium pyruvate | | |
| 2-mercaptoethanol | | |
| BSA (7.5% solution) | Thermo Scientific | 15260037 |
| N-2 Supplement (100×) | Thermo Scientific | 17502048 |
| B-27 Supplement (50×) | Thermo Scientific | 17504044 |
| PD0325901 | Stemcell Technologies | 72182 |
| CHIR99021 | Miltenyi Biotec | 130-103-926 |
| Poly-DL-ornithine hydrobromide, mol wt 3,000–15,000 | Sigma-Aldrich | P8638 |
| **Commercial assays and kits** | | |
| RNeasy MinElute Cleanup kit | Qiagen | 74204 |
| µMacs Streptavidin Kit | Miltenyi | 130-074-101 |
| Turbo DNase | Ambion | |
| NEBNext Ultra II Directional RNA Library Prep Kit for Illumina | Neb | E7420 |
| NEBNext Multiplex Oligos | Neb | E7335, E7500 |

## Ethical statement

All applicable international, national, and institutional guidelines for the care and use of animals were followed. Animal experiments were carried out under UK Home Office Project Licenses PPL 60/4424 and PB0DC8431 and were approved by the University of Edinburgh animal welfare and ethical review body.

## Gene targeting in ESCs

The bacterial artificial chromosome (BAC) bMQ420i24 containing chr11:69711833-69860134 (mm10 assembly) of the mouse genome from 129S7/SvEvBrd ES cells (Adams *et al*, 2005) was modified to introduce the R749H mutation into exon 14 of *Polr2a* using a *GalK* selection cassette as described (Warming *et al*, 2005). A ~10.3-kb region (chr11:69741333-69751734) of the *Polr2a* locus was then

retrieved into the *NotI-SpeI* region of PL253 using gap repair, and a Frt-flanked neomycin resistance cassette from plasmid PL451 introduced into intron 12 of the gap-repaired *Polr2a* clone at position chr11:69743748-69743749 as described (Liu *et al*, 2003). The resulting targeting vector was linearized with *NotI* and introduced into E14 ESCs by electroporation (Joyner, 2000). Genomic DNAs from were screened for homologous recombination by PCR. The neomycin resistance cassette was then excised by electroporation with an Flp recombinase expression plasmid to generate WT/slow heterozygous ESCs. The same targeting vector was used to target the WT *Pol2ra* allele in the WT/slow ESCs, and the neomycin resistance cassette excised using Flp recombinase in order to generate slow/slow ESCs. WT/WT, WT/slow, and slow/slow ESCs were confirmed to contain forty chromosomes by karyotyping as described (Nagy *et al*, 2009). Ion Torrent sequencing of overlapping PCR products from ESC genomic DNA encompassing a

~14-kb region around the R749H mutation (chr11:69739041-69753349) was used to confirm that the WT/slow and slow/slow ESCs contained no genomic re-arrangements or additional mutations in this region relative to the parental WT/WT ESCs. Genotyping was performed using the following forward and reverse primers: GGGACTCCATTGCAGATTC and ACTCAGTGGGTGTGAGACC.

## Mice chimera generation and breeding

In order to generate mouse chimeras, WT/slow ESCs were injected into C57BL/6 host chimeras, as previously described (Joyner, 2000). Eight male chimeras with at least 30% contribution from ESCs were identified by coat color and bred with C57BL/6 females to test for germline transmission.

## CRISPR/Cas9 gene editing in mouse zygotes

Complementary oligonucleotides targeting exon 14 of *Polr2a* were annealed and cloned into plasmid pX335 (Cong *et al*, 2013). The guide region was then amplified by PCR and paired guide RNAs synthesized by *in vitro* transcription (T7 Quick High Yield RNA Synthesis kit, NEB). Single-stranded DNA oligonucleotides (silent oligo: TCATTGAGAAGGCTCATAACAATGAGCTA

GAACCCACTCCAGGAAACACATTGAGACAAACATTTGAGAATC AAGTGAATCGTATTCTCAATGATGCTAGGGACAAAACTGGCTCCT CTGCACAGAAATCCCTCTCTGAATATAACAACTTCAAGTCTTGGT GGTGTCTGGAGCCAAGGGTTCCAAGATCAACATCTCC, slow oligo: TCATTGAGAAGGCTCATAACAAT

GAGCTAGAACCCACTCCAGGAAACACATTGAGACAAACATTTG AGAATCAAGTGAATCGTATTCTCAATGATGCTCATGACAAAACTG GCTCCTCTGCACAGAAATCCCTCTCTGAATATAACAACTTCAAGTC TATGGTGGTGTCTGGAGCCAAGGGTTCCAAGATCAACATCTCC) were synthesized by IDT. Gene editing was performed by microinjection of RNA encoding the Cas9 nickase mutant (50 ng/μl, TriLink BioTechnologies), paired guide RNAs (each at 25 ng/μl), and 150 ng/μl single-stranded DNA oligonucleotide repair template in (C57BL/6 × CBA) F2 zygotes (Crichton *et al*, 2017), and the injected zygotes were cultured overnight in KSOM for subsequent transfer to the oviduct of pseudopregnant recipient females (Joyner, 2000), or for 3 days to allow analysis of morula/blastocyst stage embryos. CRISPR/Cas9 gene editing can generate mosaic embryos (Yen *et al*, 2014), but for simplicity, embryos that were genotyped to contain both a wild-type and a mutant Polr2a allele were classified as heterozygotes. Genotyping was performed as above except for blastocysts genotyping where nested PCR was performed, using first the above forward and reverse primers, followed by second PCR using the following forward and reverse primers: GAAGGCTGGGCAGAGAAGAG and TCCGCTTGCCCTCTACATTC

## *In vitro* transcription assay

Nuclear extracts were prepared, as previously described (Folco & Reed, 2014). A DNA construct, containing CMV promoter and encoding β–globin, was linearized by restriction digest. *In vitro* transcription reactions were performed at 30°C in 25 μl reaction mixtures containing 375 ng DNA template, 1 μl 32P-UTP, 10 μl ESC nuclear extract, 10 mM ATP, CTP, GTP, 0.4 mM UTP, 3.2 mM

MgCl$_2$. Following indicated time, proteinase K was added to stop transcription. RNA was extracted and run on denaturing polyacrylamide gel and detected by Phosphorimager.

## Nascent transcription assays

Elongation rate experiments were carried out as described (Singh & Padgett, 2009). Briefly, cells were treated for 4 hr with 100 μM 5,6-dichlorobenzimidazole 1-β-d-ribofuranoside (DRB) to inhibit transcription. To restart transcription, cells were washed twice in warm PBS, and incubated with fresh medium. During 0- to 180-min incubation, at indicated times, cells were lysed directly in TRIzol and RNA was extracted according to manufacturer's recommendations. 5 μg of total RNA was reverse transcribed using random hexamers and Superscript III. Pre-mRNA levels were measured by quantitative RT–PCR using Sybr Green Master Mix and Lightcycler 480 (Bio-Rad). Primers used in the quantitative RT–PCR are available on request. Pre-mRNA levels were normalized to pre-mRNA levels at $t = 0$ min. Results depict average of three independent experiments, ± standard error.

## 4sU-DRBseq

ESCs were seeded in 15-cm plates in 2i medium. At the 80–90% confluency, cells were treated with 100 μM DRB, in three biological replicates. Following 4 h of incubation, DRB-containing media were removed, and cells were washed twice with warm PBS and placed in fresh media without DRB. 4-thiouridine (4sU) was added to medium at a final concentration of 1 mM for 10 min before each harvest. Cells were lysed directly on a plate with 5 ml of TRIzol at indicated transcription elongation time point. Total RNA was isolated as per manufacturer's instructions. Total RNA (100–200 μg) was used for biotinylation and purification of 4sU-labeled nascent RNAs. The biotinylation reaction consisted of total RNA and EZ-Link HPDP-Biotin dissolved in dimethylformamide (DMF) and was performed in labeling buffer (10 mM Tris pH 7.4, 1 mM EDTA) for 2 h with rotation at room temperature. Unbound Biotin-HPDP was removed by chloroform/isoamylalcohol (24:1) extraction in MaXtract tubes (Qiagen). RNA was precipitated with 10$^{th}$ volume of 5M NaCl and 1 volume of isopropanol. Following one wash in 80% ethanol, the RNA pellet was left to dry and resuspended in 100 μl RNase-free water. Biotinylated RNA was purified using μMacs Streptavidin kit. Specifically, 100 μl of beads per 100 μg of RNA was incubated with rotation for 15 min and then washed three times with washing buffer (100 mM Tris pH 7.5, 10 mM EDTA, 1 M NaCl, 0.1% Tween-20) at 65°C, followed by three washes at room temperature. RNA was eluted twice using 100 mM DTT and recovered using RNeasy MinElute Cleanup column (Qiagen) according to instructions. cDNA libraries were prepared using NEB Next Ultra Directional RNA Library Prep Kit according to the manufacturer's instructions. Libraries were pooled and sequenced on an Illumina HiSeq 4000 system. All reads were aligned to the mouse reference genome (mm10) using bowtie 2 aligner (Langmead *et al*, 2009), and only those reads that mapped uniquely to the genome, but not to rRNA, were considered. A genome-wide binned profile of the nascent RNA and the transcription wave end were determined using previously developed methods and published software (Fuchs *et al*, 2014, 2015).

## Cell differentiation

ESCs were tested for mycoplasma contamination. ESCs were cultured under feeder-free conditions in GMEM supplemented with 10% fetal calf serum, NEAA, β-mercaptoethanol, sodium pyruvate, L-glutamine, and 100 U/ml recombinant leukemia inhibitory factor (LIF) on gelatin-coated tissue culture plastic. Before differentiation, cells were freshly defrosted in standard medium and then passaged for 2 passages in 2i medium (1:1 Neurobasal and DMEM/F12, supplemented with 0.5× N2, 0.5× B27, 0.05% BSA, 1 μM 0325901, 3 μM CHIR99021, 2 mM L-glutamine, 0.15 mM monothioglycerol, 100 U/ml LIF).

## Neuroectodermal specification

One day prior to induction of differentiation, cells were seeded at high density in 2i medium. The following day, cells were detached using accutase, resuspended in N2B27 media (1:1 Neurobasal and DMEM/F12, supplemented with 0.5× N2, 0.5× B27, 0.1 mM β-mercaptoethanol, 0.2 mM L-glutamine), counted, and plated at the appropriated density onto either 15-cm plates or 6-well plates that have been coated with a 0.1% gelatin solution. Culture medium was changed every second day. The differentiation potential is greatly influenced by the initial plating density and was previously established to be optimal at approximately 10,000 cells per cm$^2$, which is what we observed with the differentiation of WT ESCs. On the contrary, we observed increased cell death at plating densities below 30,000 cells per cm$^2$ for slow/slow cells, suggesting decreased proliferation or compromised differentiation potential of these cells.

## Deriving NS cells

For derivation of neural stem cells at day 7 of differentiation, cultures were detached using accutase, 2–3 × 10$^6$ cells were re-plated into an uncoated T75 flask in NS expansion media, comprising DMEM/F12 medium supplemented with 2 mM L-glutamine, 0.5× N2, B27, glucose, BSA, HEPES, and 10 ng/ml of both mouse EGF (Peprotech) and human FGF-2 (Peprotech). Within 2–3 days, thousands of cell aggregates formed in suspension culture and were harvested by centrifugation at 700 rpm for 1 min. They were then re-plated onto a laminin-coated T75 flask. After few days, cell aggregates attached to the flask and outgrew with NS cell.

## Differentiation to neurons

For neuronal generation and maturation at day 7 of differentiation, cultures were detached using accutase, re-plated onto poly-l-ornithine-/laminin-coated surfaces (100 μg/ml and 10 μg/ml, respectively, Sigma-Aldrich) at 1.5–2 × 10$^4$ cells/cm$^2$ in N2B27 medium containing 0.2 mM ascorbic acid and 0.25 mM cAMP. Cells were grown for the additional 14 days, with 80% media exchange every second day.

## RNA isolation and RT–qPCR

RNA was isolated using TRIzol or RNeasy following the manufacturer's protocol. RNA was then treated with DNase (Ambion) and transcribed to cDNA using First-Strand Synthesis System from Roche. This was followed by SybrGreen detection system (Lightcycler 2× SybrGreen Mix, Roche).

## RNA purification and RNA-Seq analysis

RNA sequencing was carried out on RNA extracted from WT/WT and slow/slow ESCs, ESC-derived NPCs at day 7 of differentiation and ESC-derived neurons at day 21 of differentiation. RNA was purified using RNeasy kit from three independent differentiation experiments. RNA-seq libraries were generated from Poly(A)$^+$ mRNA using TrueSeq protocol and sequenced using the Illumina HiSeq 4000 machine (Edinburgh Genomics) to generate 75 bases, paired-end reads. Reads were mapped to the mouse (mm9) genome. AS analysis of RNA-Seq data was performed with *vast-tools* version 1 (Tapial *et al*, 2017). From the primary output, events with poor coverage or junction balance were filtered out (vast-tools quality score 3 other than SOK/OK/LOW for cassette exon [CE], microexon [MIC], and alternative 5′ or 3′ splice site [Alt5/3] events or coverage < 15 reads for intron retention [IR] events; score 4 other than OK/B1 for CE and MIC events and score 5 of < 0.05 for IR events). Differential AS was scored using vast-tool's diff module requiring p(|dPSI| > 0) > 0.05 and a point estimate of |dPSI| > 10. Gene expression was analyzed based on raw read counts per gene from *vast-tools* using the glm stream of the R package edgeR. Genes with an FDR < 0.05 were considered differentially expressed. Clustering of the samples shows very good correlation between results obtained in the three independent experiments (Appendix Fig S6).

## Networks

The GO network for the genes misregulated in cells harboring slow RNAPII was built using Enrichment Map (Merico *et al*, 2010) in Cytoscape 3.3.1 (Shannon *et al*, 2003) with the following parameters: *P*-value cutoff = 0.001; FDR Q value cutoff = 0.01; Jaccard+Overlap Combined option, with cutoff = 0.375; Combined Constant = 0.5. Enriched functional terms were obtained from g:profiler or by GSEA for GO_BP and KEGG pathways. g:profiler was employed for the analysis of GO enrichment during neuronal differentiation.

## Statistics

To determine statistical significance, unpaired *t*-tests were used to compare between two groups, unless otherwise indicated. The mean ± the standard error of the mean (SEM) is reported in the figures as indicated. Statistical significance was set at *P* < 0.05. All *in vitro* experiments were repeated three times, and several litters were analyzed animals for *in vivo* studies. Fisher's exact test was used to determine the significance in animal studies.

# Data availability

RNA-seq and 4sU-DRB-seq data generated in this study have been submitted to the NCBI Gene Expression Omnibus (GEO; http://www.ncbi.nlm.nih.gov/geo/) under accession number GSE127741.

**Expanded View** for this article is available online.

## Acknowledgements

We are grateful to Nick Hastie (MRC HGU) and Wendy Bickmore (MRC HGU) for support at the initial stages of this project. We thank Graeme Grimes (MRC HGU) for help with bioinformatics analysis and Joe Marsh (MRC HGU) for discussions about RNAPII structure. We thank Wendy Bickmore (MRC HGU) for critical reading of the manuscript. We acknowledge the contributions of David Read, who recently passed away. This work was supported by Core funding to the MRC Human Genetics Unit from the Medical Research Council (J.F.C and I.A), by a Wellcome Trust Investigator Award (Grant 095518/Z/11/Z to J.F.C) and by the Canadian Institutes of Health Research (BJB).

## Author contributions

MMM, IRA, and JFC conceived, designed, and interpreted the experiments. UB and BJB provided bioinformatics analysis and discussion of AS changes during differentiation. SA contributed to the bioinformatics analysis of RNAPII elongation rates. ARM, CJH, and FK carried out ESCs targeting. The CBS Transgenic Core performed CRISPR injections and mouse work. ARK participated in discussions. MMM, BJB, ARK, IRA, and JFC wrote the paper.

## Conflict of interest

The authors declare that they have no conflict of interest.

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
