## [Review Process File · The EMBO Journal]

A slow transcription rate causes embryonic lethality and perturbs kinetic coupling of neuronal genes

Magdalena M. Maslon, Ulrich Braunschweig, Stuart Aitken, Abigail R. Mann, Fiona Kilanowski, Chris J. Hunter, Benjamin J. Blencowe, Alberto R. Kornblihtt, Ian R. Adams & Javier F. Cáceres

Review timeline:

Submission date:	28th Nov 2018
Editorial Decision:	7th Jan 2019
Revision received:	11th Feb 2019
Editorial Decision:	1st Mar 2019
Revision received:	5th Mar 2019
Accepted:	7th Mar 2019

Editor: Karin Dumstrei

Transaction Report:

1st Editorial Decision

7th Jan 2019

Your manuscript has now been seen by three referees and their comments are provided below. As you can see the referees find the analysis interesting, but they also find that further experiments are needed to support the key conclusions. Their comments are constructive and reasonable. Should you be able to address the concerns raised then I would like to invite you to submit a revised version. I should add that it is EMBO Journal policy to allow only a single round of major revision and that it is therefore important to resolve the key concerns at this stage.

Let me know if we need to discuss any experiments further.

REFeree REPORTS:

Referee #1:

This manuscript investigates the *in vivo* connection and relevance of transcription speed and gene expression. This is a worthwhile extension of the many studies linking pol II speed and gene expression in cell lines and biochemical systems.

The authors convincingly show that ESCs that express slow polymerase are not competent for development. Therefore, the authors are limited to testing the role of transcription speed in a cultured model of development, rather than truly addressing the question of the molecular impact of slowing polymerase *in vivo*. (which would presumably require an inducible system later in development).

While not fully addressing the ultimate *in vivo* relevance, overall the data in this manuscript support the notion that polymerase speed is important for proper cellular development. However, in some places further analysis or clarity in the text would strengthen the manuscript.

- 1) The abstract is written to imply that neural development is specifically sensitive to elongation speed. However the manuscript ONLY looks at neural development. This should be clarified.
- 2) Are the genes that exhibit speed-dependent AS also those that show expression differences? The authors have a paragraph saying "no" to this question, but no data is shown.
- 3) Are the differences in AS in ESC, NPC and neurons due to differential gene expression? In other words are the genes that only show speed-dependent AS in neurons only expressed in neurons, or are there any genes that are expressed in all stages of development but sensitive to Pol II speed only in one condition and not another.
- 4) EV Fig 5 is missing labels, rendering it difficult to understand.

Referee #2:

Pre-mRNA splicing is an essential process that largely occurs co-transcriptionally and allows for regulation of gene expression via alternative splicing (AS). Although AS is primarily regulated by RNA binding proteins, various lines of evidence have indicated that altered RNA Pol II kinetics can influence AS patterns via kinetic coupling. Many of these experiments have used an RNA Pol II large subunit mutation that slows the elongation rate, usually in conjunction with an alpha-amanatin resistance mutation that allows selection of transfected cells that are dependent on the "slow" (or "fast") RNA Polymerase.

The Cáceres lab now reports an attempt to test the consequences of the "slow" polymerase mutation (R749H) in vivo using mouse models without the need for alpha-amanatin resistance. Two independent attempts to introduce R749H into mice failed to achieve germline transmission, even as heterozygotes, indicating strong selection against this mutation. This in itself is a significant observation.

The authors then proceeded to analyse the effects of R749H in mES cells during differentiation to neural precursors, neural stem cells and neurons in vitro. A defect was found in NSCs, but direct neuronal differentiation from NPCs was viable allowing transcriptome profiling of WT and slow/slow cells in ESCs, NPCs and neurons. Alterations in RNA abundance and AS were observed, with long neuronal genes particularly affected.

Overall, I find this to be an interesting manuscript reporting significant new findings on the coupling of transcription and splicing. However, there are some points in the text and figures that need attention.

Major comments

1. In view of the fact that all the subsequent mRNA-Seq data are based on the direct differentiation route to neurons via poly ornithine/laminin, it would be appropriate to include more data in Fig 4 on the characterization of these neurons in addition to the Tuj1 staining shown in Fig 4E. The data in EV3 suggest not only that Syn1 levels are much lower, but also that Map2 is higher and the overall cellular organization looks quite different.

2. p13 continued. "...thus, these results underscore the predominant role of kinetic coupling as differentiation progresses." Also on p17, last sentence and Discussion section.

An alternative explanation might be that some of the changes seen in NPCs and neurons are indirect consequences of changes initially caused in ESCs or NPCs (e.g. is expression of Rbfox and/or Nova affected - see below). This should be discussed, and in general the manuscript could be improved by taking a more critical approach to the kinetic coupling model.

3. P14-15. The statement that binding of RbFOX and NOVA downstream of exons increases their skipping is incorrect. Both protein families are generally associated with activation of splicing from downstream locations. The observation is that exons showing more skipping with slow polymerase

in neurons are associated with RbFOX and NOVA motifs in locations where they would usually activate. It would also be appropriate to draw attention in the text to the strong association of slow-Pol upregulated exons with NOVA sites in a repressive upstream location.

4. Related to the preceding point, were Nova and/or Rbfox proteins themselves regulated at the RNA abundance or AS levels? This should be stated in the text. If either are down-regulated, it would provide a neat explanation, independent of kinetic coupling, for altered AS of these groups of exons. Conversely, if Rbfox and Nova expression are not affected, it could indicate an interesting functional connection between Rbfox/Nova regulation and kinetic coupling.

5. P15. The observation of a small overlap between genes regulated at the splicing and RNA abundance levels is quite important. It would be useful to have Venn diagrams (or some other representation) showing the overlap of genes regulated at each level - perhaps separated by AS event type, given the possibility of NMD coupling with intron retention.

6. Figure 5A. The data on AS changes could be summarized more informatively to include information on the magnitude of splicing changes (delta PSI) as well as the raw numbers of up and down regulated events.

Minor points

7. Fig EV2C. Quantification of the correlation should be given.

8. Fig 3D. Given the bimodality of WT/WT in Fig 3C, it would be appropriate to show a violin plot rather than a boxplot.

9. Fig 4A. It would help to indicate on the Figure the stages at which different markers (Sox1, Pax6, Nestin, Tuj1, Map1, Syn1) are expressed.

10. The whole manuscript is predicated on the fact that R749H results in a "slow" RNA polymerase. The mutant was originally characterized before high resolution RNA Pol structures were available. It would be useful to include a couple of sentences - either in Introduction or Discussion - on the molecular consequences of R749H on Pol II function in the context of high resolution RNA Pol II structures. Do the structures explain the elongation defect well? Is it possible that the mutation has other effects on Pol II function, in addition to the elongation defect, that might contribute to the observed phenotypes? If this has been discussed elsewhere, a citation would be sufficient.

11. The section describing the ESC, NPC, NSC and neuronal culture needs some clarification in the text (p11-12), as well as in Figs 4 and EV3. Fig EV3 gives no indication of what cells are shown (by which route were they derived), and the histograms in Fig EV3B need labeling to show WT and slow. Since Tuj1 neurons were seen in the NSC cultures, the subsequent text (further down p12) on Tuj1 positive neurons needs to emphasise that they were obtained by the poly-ornithine/laminin route (But see major point 1).

12. Page 12, line 4. It was not immediately clear which data (publicly available or new data) was used for GO analyses. The mRNA-Seq experiment is not described until p13, but it does not seem to include aggregates. This needs clarification.

13. p13 last sentence "...extent of splicing changes was much more pronounced in NPCs...and neurons...in comparison to ESCs."

P14 line 2 "Importantly, the extent of AS events observed in the different stages of neuronal differentiation is comparable". Presumably the second sentence refers only to NPC vs neurons? Might be better to state this explicitly otherwise the two sentences seem to contradict each other.

14. Fig6A. Better contrast is needed between CC, BP and MF (BP and MF were indistinguishable on my printed copy).

15. Fig 6B needs an indication of which comparisons are significantly different.

16. In other Figures there is no indication of the number of replicates e.g. Fig 5B, EV1c, EV 3b

Referee #3:

The authors of this paper attempt to elucidate the role of transcriptional elongation rate in regulating splicing fidelity. Based on previous studies, it is known that an optimal rate of transcriptional elongation is necessary for co-transcriptional splicing activity in both yeast and mammalian cells. The mechanism controlling transcriptional coupling to alternative splicing (AS) *in vivo* remains undefined and the authors sought to address two main objectives: i) determine the effects of altered transcriptional elongation rate on gene expression and AS during mammalian development and ii) identify the consequences of tissue/organism phenotype when AS is misregulated.

In order to answer these questions, the authors generated a slow RNA polymerase II (RNAPII) mutation in embryonic stem cells (ESCs). The first attempt at introducing a mutant RNAPII allele by homologous recombination was unsuccessful in producing germline transmission in mouse chimeras. To investigate the developmental effects of a slow RNAPII allele further, the investigators generated mutant RNAPII embryos using CRISPR/Cas9. While the slow RNAPII mutation was embryonic lethal in ESCs both heterozygous and homozygous for the allele, the authors were able to show that the mutation results in a statistically significant decreased transcriptional elongation rate in mouse ESCs by DRB treatment.

The authors next looked at the effects of altered transcription elongation rates on splicing during neural differentiation. The authors claimed that the slow mutation in RNAPII causes defective self-renewal of neural stem cells rather than issues with neuronal differentiation. They also found differences in AS in neural progenitor cells (NPCs) and neurons differentiated from slow/slow ESCs when compared to WT. This effect was particularly enhanced in long genes expressed during neuronal differentiation, and the investigators suggest this is the result of improper kinetic coupling. Taken together, the authors indicate that an appropriate transcriptional elongation rate is required for normal gene expression during development.

This manuscript sets out to address interesting and important questions, and does so through an impressive amount of work. The authors provide evidence that the slow RNAPII allele is embryonic lethal and results in decreased elongation rates in ESCs, their experimental evidence in support of kinetic coupling during neural differentiation is less strong. However, there are a number of critical weaknesses with the paper. In particular, the authors claim that transcriptional elongation rate influences alternative splicing, but fail to provide any statistics or details on experiments to reproduce AS changes in WT and slow/slow ESCs, NPCs, and neurons. In the results section, the authors also suggest that kinetic coupling is enhanced in neurons, but do not provide any comparison to other cell types, let alone ESCs.

The following suggestions may improve the effectiveness of the paper:

(1) One critical weakness with the current manuscript is that many details on the extent and quality of replication experiments are lacking. The authors should provide details on the number of replicates that were generated in both mRNA-seq, 4sU-seq, and other genomic experiments, what the extent to which experiments were concordant to the main conclusions.

While the general idea that RNAPII mutants have a slower elongation rate is convincing as presented, there are many details that could be altered if the experiment were replicated (for example, are there really a group of genes that increases elongation rates as claimed, or is this just due to statistical fluctuations?!).

More importantly, to what extent are alternative splicing changes reproducible between mRNA-seq replicates? The individual gene PCR/ bioanalyzer validation experiments move toward this, but appear to have been completed once, and there is no indication on whether/ how many genes were selected for validation but failed in this test. The authors should show replication.

(2) Report the direct effects of AS on gene expression during NSC maintenance.

(3) The authors claim that kinetic coupling is enhanced in neurons due to increased chromatin compaction, but there is no direct evidence of either increased chromatin compaction, or that it is causally related to kinetic changes. Please either provide some evidence (i.e. immunofluorescence,

ATAC-seq, DNase-seq) that the changes in nucleosome occupancy during neuronal differentiation affects AS levels in NPCs and neurons, or weaken this statement substantially.

(4) Are the expression levels and splicing activity of long genes in ESCs similarly affected by a slow RNAPII as neurons or is this result specific to neurodevelopment? ESCs should be added to Fig. 6 panel C.

(5) I think it might help set the stage for readers to present the section "The R749H mutation decreases the transcription elongation rate in mouse ESCs" as the second section of the paper, as it probably will not be 100% clear to readers that previous studies of slower elongation rates with this mutation in *Drosophila* will hold up in mice. It is completely up to the authors whether or not to implement this suggestion.

1st Revision - authors' response

11th Feb 2019

We are grateful to all the referees for their constructive and rigorous input, which has helped to strengthen our conclusions. The manuscript has been revised in response to the comments of all reviewers with the addition of some new data.

We are including below a "List of major changes" as well as a detailed Point-by-Point response to referees, where we have attempted to address each one of the comments raised by the reviewers. Finally, a 'marked' version of the Manuscript text has been included with the Supplementary files.

Paste in PbP.

2nd Editorial Decision

1st Mar 2019

Thank you for sending us the revised manuscript. Your revision has now been seen by the three referees and their comments are provided below.

As you can see, the referees appreciate the introduced revisions and support publication here. Please take a look at the minor point raised by referee #2 and respond.

Besides this, we just need the following editorial comments taken care of. You can submit the revised version using the link below. Let me know if you have any questions

REFEREE REPORTS:

Referee #1:

The authors have addressed all of my concerns with additional data and modifications to the text.

Referee #2:

The authors have engaged constructively with reviewer comments. I have no further serious issues to raise.

Minor point. Figure 5B shows AS events validated by RT-PCR. No indications of statistical significance are given, but the error bars for *Pbrm1* suggest that there is no significant change. Perhaps a better example could be shown?

Referee #3:

The authors have addressed my comments.

2nd Revision - authors' response

enter date

The authors performed all requested editorial changes.

Corresponding Author Name: Javier F. Caceres and Ian R. Adams

Journal Submitted to: Embo Journal

Manuscript Number: EMBOJ-2018-101244